# Tribological Behavior of Sulfonated Polyether Ether Ketone with Three Different Chemical Structures under Water Lubrication

**DOI:** 10.3390/polym16070998

**Published:** 2024-04-05

**Authors:** Xiaozhi Chen, Tao Hu, Wei Wu, Xiaohong Yi, Fenghua Li, Chenhui Zhang

**Affiliations:** 1School of Metallurgy, Northeastern University, Shenyang 110819, China; 2101617@stu.neu.edu.cn; 2Engineering Center for Superlubricity, Jihua Laboratory, Foshan 528200, China; hutao@jihualab.ac.cn (T.H.); scutw.wei@gmail.com (W.W.); chzhang@tsinghua.edu.cn (C.Z.); 3State Key Laboratory of Tribology, Tsinghua University, Beijing 100084, China

**Keywords:** sulfonated polyether ether ketone (SPEEK), chemical structures, tribological properties, hydration lubrication

## Abstract

With the development of the shipbuilding industry, it is necessary to improve tribological properties of polyether ether ketone (PEEK) as a water-lubricated bearing material. In this study, the sulfonated PEEK (SPEEK) with three distinct chemical structures was synthesized through direct sulfonated polymerization, and high fault tolerance and a controllable sulfonation degree ensured the batch stability. The tribological and mechanical properties of SPEEK with varying side groups (methyl and tert-butyl) and rigid segments (biphenyl) were compared after sintering in a vacuum furnace. Compared to the as-made PEEK, as the highly electronegative sulfonic acid group enhanced the hydration lubrication, the friction coefficient and wear rate of SPEEK were significantly reduced by 30% and 50% at least without affecting the mechanical properties. And lower steric hindrance and entanglement between molecular chains were proposed to be partially responsible for the lowest friction behavior of SPEEK with methyl side groups, making it a promising and competitive option for water-lubricated bearings.

## 1. Introduction

Friction and wear are ubiquitous phenomena in both everyday human life and industrial production. They serve as the primary mechanisms and sources of mechanical energy consumption and failure [1,2,3]. In light of the ongoing advancements and growth in human society, economy, and science, along with the pressing demand for the advancement of the manufacturing industry and ecological civilization, tribology is confronted with numerous new challenges. Particularly in bearings, addressing the issues of limited lifespan and excessive energy consumption, as well as finding ways to conserve resources and promote environmental sustainability without compromising material performance, has emerged as a pressing concern that requires immediate attention. Green tribology has emerged as a prominent area of focus within the field of contemporary tribology [4,5,6]. Water-lubricated bearings based on water-based lubrication have attracted significant attention among scholars due to their cost-effectiveness, environmental friendliness, and easy maintenance [7,8,9]. Water exhibits numerous distinctive properties. The enduring fluidity of water within confined thin films was discovered, which was pertinent to friction and lubrication processes [10]. However, the limited viscosity and carrying capacity of pure water hinder the formation of a comprehensive lubricating film and hydrodynamic lubrication in water-lubricated bearings operating under low speed and heavy load conditions [11,12,13]. Furthermore, it has been observed that the utilization of water-based lubrication can lead to a corrosive impact on mechanical components composed of steel and its alloys [9]. The aforementioned deficiencies have a significant impact on the dependability of subaquatic equipment and severely restrict the feasible utilization of water-lubricated bearings. Higher demands are also imposed on the water-lubricated bearing materials.

As the primary materials for the friction pair in water-lubricated bearings, polymer materials have been increasingly replacing traditional water-lubricated bearing materials, due to their excellent vibration absorption properties and high wear resistance [14,15,16]. Based on the concept of hydration lubrication, hydrogels [17] and molecular brushes [18] with polar or charged functional groups designed to enhance lubrication have been significantly reported. Hydration lubrication describes the lubrication mechanism through hydrated ions trapped between charged surfaces [19,20,21]. These hydrated ions with hydrated shells or sheaths are probably very strongly attached and rapidly relaxing at the same time. Simultaneously, the formation of the hydration layer by hydrated ions effectively serves as a barrier to prevent direct contact between the sliding surfaces. This phenomenon results in a large carrying ability and sustains ultra-low shearing resistance between sliding surfaces [22]. Nevertheless, the investigation of hard friction pair materials has been relatively limited in comparison to soft friction materials.

Polyether ether ketone (PEEK) has been widely utilized as a friction pair material in various fields owing to its exceptional mechanical strength, favorable chemical inertness, and excellent friction and wear characteristics [23,24,25]. The enhancement of PEEK’s tribological properties has garnered significant attention in the academic community. However, the application of negative electric modification to PEEK in the context of hydration lubrication has been scarcely documented. Negative electric modification of PEEK could improve the surface hydrophilicity, which was conducive to the formation of aqueous film and achieved friction-reduction performance and anti-wear property. As a new material for water-lubricated bearings, the sulfonation of PEEK to enhance the water-lubricating performance was first reported by our group. Sulfonation modification is a common modification method for PEEK on account of its simplicity and effectiveness. The sulfonated PEEK (SPEEK) can be synthesized through sulfonation or polymerization [26,27,28]. The former utilizes sulfonation reagents (concentrated sulfuric acid and chlorosulfonic acid) for direct sulfonation, while the latter utilizes sulfonated monomers. Yuan et al. [29] successfully obtained SPEEK with ultra-low friction coefficient through sulfonation using chlorosulfonic acid and sintering via a sparking plasma sintering (SPS). Wu et al. synthesized a novel bisphenol-A (BPA)-type sulfonated PEEK and compounded it with 10 wt% carbon fiber, which exhibited a friction coefficient as low as 0.009 and a wear rate of grade 10^−8^ when a 3 wt% NaCl solution was used as the lubricant at a low sliding speed (0.10 m/s) [30]. In comparison, the SPEEK’s degree of sulfonation was controllable and repeatable using sulfonated monomer polymerization, and the sulfonic acid groups on the benzene rings had better chemical stability and strong hydrolysis resistance [27,28].

The microscopic influence of chemical structure on tribological properties has been a research hotspot. However, there are few studies on thermoplastic materials with complex sintering processes. In this study, the SPEEK powders with three distinct structures were polymerized through three kinds of monomers with diverse side groups and rigid segments. The SPEEK were sintered in a vacuum furnace, and the tribological properties were studied for the first time. Combining the analysis of thermal properties and static contact angles, the effect of polar $–{\mathrm{SO}}_{3}^{-}$ groups and chemical structures on the friction coefficient and wear rate was systematically assessed. The underlying mechanism responsible for these properties was elucidated, which highlighted the importance of chemical structures to friction properties. It was beneficial for the design of the water-lubricating material based on hydration lubrication effect.

## 2. Materials and Methods

### 2.1. Materials

The reagents included disodium 3,3′-disulfonate-4,4′-difluorobenzophenone (Alfa Chemical, Zhengzhou, China), 4,4′-difluorobenzophenone (Macklin, Shanghai, China), 2-methylhydroquinone (Bidepharm, Shanghai, China), tert-butylhydroquinone (Meryer, Shanghai, China), and 3,3′,5,5′-tetramethylbiphenyl-4,4′-diol (Bidepharm, Shanghai, China). All the additional reagents and solvents (potassium carbonate, tetrahydrothiophene 1,1-dioxide, toluene, ethanol) were purchased commercially and utilized without further purification.

### 2.2. Synthesis of the SPEEK Powders

Based on the chemical equation depicted in Figure 1a, SPEEK was polymerized through an aromatic nucleophilic substitution reaction [31,32,33]. First, monomer 1 (disodium 3,3′-disulfonate-4,4′-difluorobenzophenone), monomer 2 (4,4′-difluorobenzophenone), monomer 3 (2-methylhydroquinone, tert-butylhydroquinone, 3,3′,5,5′-tetrameth-ylbiphenyl-4,4′-diol), and basic catalyst potassium carbonate (K_2_CO_3_, 27.6 g) were added in a three-necked flask equipped with an electric stirrer, airway, and water separator. The distribution of the three monomers in the polymerization reaction was shown in Table 1. The as-made pure PEEKs were without sulfonated monomer 1. Herein, SPEEK used 2-methylhydroquinone as monomer 3 was designated as SMPEEK; SPEEK used tert-butylhydroquinone as monomer 3 was designated as STPEEK; SPEEK used 3,3′,5,5′-tetramethylbiphenyl-4,4′-diol as monomer 3 was designated as STDPEEK. Then, 150 mL of the organic solvent tetrahydrothiophene 1,1-dioxide (TMS) and 50 mL of the water-carrying agent toluene (PhMe) were introduced. The mixture was agitated at 300 rpm. The polymerization reaction first reacted at 140 °C for 4 h, after the removal of water from the reaction mixture by azeotropic distillation, and the reaction temperature was subsequently increased to 200 °C and kept for 6 h. A continuous flow of nitrogen was passed abidingly through the airway during the entire reaction.

At the end of polymerization, the heated mixture was poured into 50 wt% ethanol aqueous solution. The products were crushed in a grinder and washed repeatedly with deionized water and ethanol to remove inorganic salts and solvents. Subsequently, the SPEEK products were dried at 60 °C for 12 h, and then, we obtained SPEEK powders.

### 2.3. Sintering Procedure

SPEEK powders were sintered through a vacuum furnace (VHPgr-20/20/30-2300, Haoyue Technology, Shanghai, China). The sintering procedure was shown in Figure 1b. In the initial phase, the temperature was raised to 250 °C at a heating rate of 5 °C/min. The temperature was maintained for 10 min until stable. Subsequently, the temperature was raised from 250 to 340 °C at a heating rate of 2.5 °C/min. It was maintained at 340 °C for 3 h to complete melting. The cooling rate was consistently maintained at 0.45 °C/min. The pressure control program was initiated at the onset of the cooling stage, whereby the pressure was gradually increased from 0 to 22 MPa within 1 min and sustained until the completion of sintering.

### 2.4. Characterization

The ^1^H NMR spectra were obtained via an Avance Neo 600 spectrometer (Bruker, Karlsruhe, Germany). The Fourier transform infrared (FTIR) spectroscopy was obtained via a Nicolet iS 10 (Thermo Fisher Scientific, Boston, MA, USA). The chemical composition of the SPEEK powders was analyzed via X-ray photoelectron (XPS) spectroscopy (Escalab Xi+, Thermo Fisher Scientific, Boston, MA, USA). The actual sulfonation degree was analyzed by a carbon sulfur analyzer (Elementrac CS-d, Eltra, Berlin, Germany). The relative molecular mass was characterized via GPC Gel Permeation Chromatography (PL-GPC50, Agilent, CA, USA). The particle size distribution was obtained via a laser particle size analyzer (BT-9300S, Better, Liaoning, China). The thermal properties of the SPEEK powders were identified via differential scanning calorimetry (DSC 204 HP Phoenix^®^, Netzsch, Bavaria, Germany) and thermal gravimetric analysis (TGA 55, Waters TA, New Castle, DE, USA). The heating rate of DSC and TGA was 10 °C/min in N_2_ flow. Characterization tests were performed on different batches of SPEEK to ensure reproducibility.

For the sintered SPEEK specimens, the SPEEK specimens’ surface wettability was analyzed via the contact-angle-measuring instrument (OCA25, Dataphysics, Stuttgart, Germany). The contact angles were measured five times and then averaged. The surface morphology and electron composition were observed via the scanning electron microscope (SEM) and X-ray energy-dispersive (EDS) spectrometry (Verios 5 UC, Thermo Fisher Scientific, Boston, MA, USA) after wear.

### 2.5. Tribological and Mechanical Tests

In this study, the tribological properties of the SPEEK and synthetically pure PEEK specimens were evaluated by the friction coefficient (COF) and wear rate. All tests were performed three times to ensure accuracy. Prior to conducting the tests, the specimens’ surfaces were grounded and polished to eliminate impurities and guarantee surface flatness and smoothness. The surface roughness was within 40 nm. The COFs were measured using a rotating tribometer (UMT-Tribolab, Bruker, Karlsruhe, Germany). The Si_3_N_4_ balls (diameter = 10 mm) were employed as friction pair materials. The Si_3_N_4_ ball exhibits a negative surface charge, thereby facilitating hydration lubrication [34,35,36]. During the tests, the specimens were securely attached to a disc rotated by an electric motor, and “point-surface” contact was made with the Si_3_N_4_ ball fixed to the force sensors (Figure 1c). The tests were carried out under a normal load of 15 N (the average pressure in the contact area was 170 MPa as calculated by the Hertz contact equation) and the sliding linear velocity of 50 mm/s. The 3 wt% NaCl aqueous solution was utilized as a lubricant dropping on the specimen surfaces (approximately 3 mL). In addition, the COF at various sliding linear velocities (50, 150, and 250 mm/s) and different loads (15, 25, and 35 N) were measured.

The wear rate was employed to assess the wear resistance of the SPEEK and pure PEEK specimens. After rotating friction tests at 15 N and 50 mm/s for 60 min, the wear rate was characterized by the white light interferometer (Contour GT-X, Bruker, Karlsruhe, Germany), and the wear rate of each friction pair was calculated using Equation (1):*γ* = Δ*V*/Σ*W*,(1)

Here, *γ* represents the wear rate. Δ*V* represents wear volume, which was obtained by integrating the worn cross section. Σ*W* represents the cumulative frictional work [2].

Mechanical properties were essential when the specimens as engineering materials. The primary factor taken into account was the ability to resist compression and deformation when the SPEEK was utilized as a water-lubricated bearing material. The data were the average values through multiple measurements (three times). The Young’s modulus was analyzed via a nanoindentation instrument (STEP 500 NHT^3^, Anton Paar, Graz, Austria). The load was 15 mN. Shore hardness was characterized by a shore hardness tester. Compressive strength was analyzed via a universal testing machine (CTM-E200, Quanli Test, Jinan, China). The diameter and thickness of the specimens were 20 mm and 6 mm, respectively. The test speed was fixed at 2 mm/min.

## 3. Results and Discussion

### 3.1. Analysis of the SPEEK Powders

The electron-donating groups in monomer 3 increased the electron cloud density and reactivity, thereby facilitating smooth polymerization at 200 °C. The chemical skeleton was structurally characterized by ^1^H NMR and FTIR. As shown in Figure 2, the chemical shifts observed in the spectrum corresponded to the extranuclear chemical surroundings of the hydrogen atoms. Chemical shifts of the hydrogen atoms on the main chain of benzene rings were predominantly between 6.8 and 7.9 ppm (approximately), and those on the side groups (methyl or tert-butyl) were 2.21, 1.39, and 2.20, respectively. The integral area of spectral peaks was indicative of the quantity of hydrogen atoms in the anticipated synthetic structure. Each peak exhibited a good attribution and no additional impurity peaks. Combined with FTIR, as shown in Figure 3, the absorption peaks observed at 1080 cm^−1^ and 1655 cm^−1^ corresponded to the ether bond and carbonyl, respectively. The stretching vibration absorption of O=S=O in the sulfonic acid group was at 1025 cm^−1^. It was confirmed that the polymerization reaction proceeded smoothly. The multiple absorption peaks of CH_3_ within the range of 2800~3100 cm^−1^ were caused by the rotational isomer. In addition, there were no characteristic peaks within the range of 1140~1110 cm^−1^. It was demonstrated that the copolymerization of sulfonated monomers could eliminate the degradation and crosslinking reactions, thereby ensuring a better control of the polymer properties.

The presence of S element in the SPEEK powders was observed through XPS qualitative analysis and quantitative analysis using a carbon and sulfur analyzer. As shown in Figure 4c,f,i, a distinct peak was clearly observed at 168 eV in the fine spectrum of the S 2p core-level region and could be attributed to the –${\mathrm{SO}}_{3}^{-}$ groups. The different intensities of the S 2p peak were attributed to the reaction degree. In addition, only the C=O peak (approximately 531 eV) and the C-O peak (approximately 533 eV) were observed simultaneously in the fine spectrum of O 1s core-level region (Figure 4b,e,h), and the C-O peak exhibited a greater intensity than the C=O peak. It was consistent with the anticipated chemical structures and further confirmed that the polymerization reaction proceeded smoothly. The actual degree of sulfonation can be calculated using the following Equation (2) [37]:Sulfonation degree (*D*_s_) = *S*_e_ *C*_t_/*S*_t_ *C*_e_ × 100%,(2)
where *S*_e_ and *C*_e_ are the sulfur and carbon contents based on the carbon and sulfur analyzer, respectively, and *S*_t_ and *C*_t_ indicate the content of the theoretical sulfur and carbon, respectively, when there are two –${\mathrm{SO}}_{3}^{-}$ groups per repeating unit.

According to the calculations, the actual sulfonation degrees of the SMPEEK, STPEEK, and STDPEEK powders were 1.4, 2.6, and 2.5, respectively (Table 2). Excessive sulfonation could significantly compromise the mechanical properties. It was necessary for SPEEK to have certain mechanical properties and enhanced tribological properties. The number-average molecular weight (*M*_n_) and mass average molar mass (*M*_w_) were around six to twenty thousand (Table 2). In general, tert-butyl exhibits a higher electron-donating ability than methyl, thereby promoting the formation and stability of the intermediates. The enhancement of reactivity facilitates polymerization. Laser particle size analyzer analysis showed that the median particle (*D*_50_) size was within 60 μm, which was beneficial for interfacial bonding during the sintering process.

### 3.2. Thermal Properties Analysis

The thermal properties were critical for the thermoforming process. The sintered polymers had better mechanical properties [38]. The sintering process employed a temperature regime (i.e., heating and cooling rates, sintering temperatures, etc.), which was primarily determined by the thermal properties of SPEEK. The glass transition temperature is the macroscopic embodiment of the transition of polymer chain motion form. As shown in Figure 5a,c,e, the electron-donating side groups resulted in amorphous polymers, and no endothermic or exothermic peak was observed up to 340 °C [26,30]. The arrow indicated the direction of heating. The glass transition temperatures (*T*_g_) of SMPEEK and STPEEK were 156.2 and 172.4 °C, respectively.

The identical main-chain structures resulted in similar thermal properties. Combined with the corresponding as-made pure PEEK, it was observed that a minor proportion of the –${\mathrm{SO}}_{3}^{-}$ groups had negligible impact on *T*_g_, while electron-donating side groups emerged as the primary factors affecting *T*_g_. The tert-butyl groups exhibited a larger volume than methyl groups, resulting in increased hindrance for internal rotation within polymer chains [39]. In contrast, the *T*_g_ of TDPEEK reached up to 261.9 °C. It was explained that the incorporation of rigid biphenyl significantly enhanced steric hindrance, resulting in a deterioration in the movement ability of the molecular chains [40]. The polar –${\mathrm{SO}}_{3}^{-}$ groups enhanced the entanglement between molecular chains, thereby elevating the *T*_g_ of the STDPEEK.

TGA and DTG (derivative thermogravimetry) were shown in Figure 5b,d,f. The thermal stability was influenced by the volume of side groups and molecular backbone structures. The corresponding decomposition temperatures (5% weight loss) of SMPEEK, STPEEK, and STDPEEK were 432, 486, and 460 °C, respectively. Combined with DTG, it was evident that SPEEK exhibited two thermogravimetric platforms (at the five-pointed star), whereas no such phenomenon was observed in the synthetic pure PEEK. Due to the significant electron-absorbing effect, the intermolecular bonding between the –${\mathrm{SO}}_{3}^{-}$ groups and the molecular backbone was weaker and was initially disintegrated when the temperature increased. In addition, the –${\mathrm{SO}}_{3}^{-}$ groups will lead to the weakening of the ether bonds and ketone bonds [30,41]. In our experiment, TGA and DTG indicated the absence of weight loss up to 340 °C. SPEEK had a good thermal stability and showed no thermal decomposition under the sintering temperature of 340 °C.

### 3.3. Tribology and Mechanical Analysis

The distinct chemical structures led to the difference in tribological properties and mechanical properties. The basic principle and characteristics of tribological properties were studied from the perspective of chemical structure. As shown in Figure 6a,b, the COF between the Si_3_N_4_ balls and the SMPEEK, STPEEK, and STDPEEK discs were 0.044, 0.053, and 0.064, respectively. The SMPEEK exhibited the lowest value, followed by STPEEK, while STDPEEK demonstrated the highest value. In comparison, the COF of MPEEK, TPEEK, and TDPEEK were 0.085, 0.088, and 0.092, respectively. The SPEEK specimens exhibited a noticeably lower friction coefficient than synthetic PEEK specimens.

The abrasion produced by the wear directly affects the surface roughness, which causes the fluctuation in COF in the stable stage. The electron-donating side groups had a similar impact on the friction coefficient. Figure 7a,b showed the wear surface topography and its profile curves along the direction perpendicular to the speed after rotating friction tests.

The wear marks were shallow, and a part of the wear debris was accumulated on the surface. The depth of the wear marks was within 0.5 μm. The wear rate data using formulae are shown in Figure 7c. The wear rates of SMPEEK, STPEEK, and STDPEEK were 6.2 × 10^−8^ mm^3^/(N m), 8.4 × 10^−8^ mm^3^/(N m), and 4.7 × 10^−8^ mm^3^/(N m), respectively. STDPEEK exhibited the lowest wear rate, followed by SMPEEK, and STPEEK demonstrated the highest wear rate. Correspondingly, the wear rates of MPEEK, TPEEK, and TDPEEK were 1.5 × 10^−7^ mm^3^/(N m), 1.7 × 10^−7^ mm^3^/(N m), and 1.2 × 10^−7^ mm^3^/(N m), respectively.

From the above data, it was clearly demonstrated that the polar –${\mathrm{SO}}_{3}^{-}$ groups played a significant role in anti-friction and anti-wear properties. Here, the impact of the –${\mathrm{SO}}_{3}^{-}$ groups on the SPEEK surface properties was elucidated by the static contact angle. As shown in Figure 8a, the static water contact angles of the MPEEK, TPEEK, and TDPEEK specimens were 88.5°, 89.8°, and 93.2°, respectively. The gradual increase in the contact angle was attributed to the increase in the proportion of the hydrophobic (methyl and tert-butyl) side groups. The static water contact angles of SMPEEK, STPEEK, and STDPEEK specimens were 79.4°, 85.1°, and 87.3°, respectively. The high electronegativity –${\mathrm{SO}}_{3}^{-}$ groups could attract positive charges in the water molecules, thereby increasing the adsorption capacity to the water molecules on the SPEEK specimen surface. The hydrophilicity of the specimen surfaces was enhanced, which was more conducive to hydration lubrication in forming a water film. In a neutral lubricant, the sulfonic acid groups on the surface mostly existed as –${\mathrm{SO}}_{3}^{-}$ [30], as shown in Figure 8b, and the hydration number of Na^+^ first hydration shell was 5 in the 3 wt% NaCl aqueous solution [42]. When the Si_3_N_4_ balls and the SPEEK specimens were rubbed under 3 wt% NaCl aqueous solution, the –${\mathrm{SO}}_{3}^{-}$ on the SPEEK polymer chains absorbed the hydrated Na^+^ cations through electrostatic interactions. Thus, an enhanced hydration lubricating layer was formed by hydrated Na^+^ cations, which effectively inhibited the direct contact between the surfaces of friction pairs and resulted in a significantly low sliding resistance.

The stability and repeatability of the COF were verified by varying the sliding speed and normal load in the rotating test. At a sliding speed of 50 mm/s, it was observed that the COF exhibited a gradual decrease with the increasing load, albeit to a limited extent (Figure 6c). It was observed that the lubrication state was not altered. This phenomenon could be explained by the elastic deformation of the slight convex bodies on the friction surface of the SPEEK specimens [25]. SPEEK under normal load followed viscoelastic behavior. The relation between the friction coefficient and the normal load was presented using Equation (3):*f* = *K N*^(*n*−1)^,(3)

Here, *f* represents friction coefficient, *N* represents normal load, *K* and *n* are constants, and the value of *n* is 2/3-1 [43,44]. Therefore, the COF decreased with increasing load. At a normal load of 15 N, the COF decreased as the sliding speed increased (Figure 6d). The COF of SMPEEK, STPEEK, and STDPEEK decreased significantly at a sliding speed of 150 mm/s and was 0.029, 0.034, and 0.053, respectively. The lubrication state was potentially classified as mixed lubrication. The ratio between the thickness of the fluid lubrication film and the surface roughness of the SPEEK specimens gradually increased, which were beneficial to the formation of hydrodynamic lubrication [25,43,45].

The wear mechanism was analyzed through the micromorphology of the wear surface after a rotating test. Figure 9 showed the microscopic morphology and elemental analysis of the SPEEK specimens after rotating at 15 N and 50 mm/s for 60 min. The primary composition of the wear debris on the wear surface was C, O, Na, and S. The $–{\mathrm{SO}}_{3}^{-}$ groups had not decomposed during sintering process. The wear mechanism was predominantly adhesive wear, but the wear degree of the three chemical structures was variable. The SMPEEK wear surface appeared as striped furrows, indicating that a little wear debris was generated during friction. The wear debris in the lubricating medium resulted in a lower shear strength. The STPEEK wear surface exhibited flaky wear debris. This exacerbated adhesive wear and led to an increase in the wear rate. STDPEEK without the spalling phenomenon only appears like slight convex bodies. This was attributed to the increased entanglement between the molecular chains, which resulted in a higher resistance to separation.

The SPEEK’s mechanical properties could serve as an objective measure to distinguish the variations in chemical structures. Table 3 showed the related mechanical performance parameters of synthetizing pure PEEK and SPEEK specimens. The standard deviations were small, indicating that SPEEK mechanical properties had good stability and reliability. Due to the influence of the ionomer effect [46], the introduction of $–{\mathrm{SO}}_{3}^{-}$ groups enhanced the polymer polarity and the interaction force between the molecular chains. The polymer molecules were arranged more closely together, which made their comprehensive mechanical properties superior to the pure specimens, albeit to a limited extent. Comparatively speaking, the difference in chemical structures had a significant influence on mechanical properties. The SMPEEK specimens exhibited more considerable Young’s modulus and shore hardness. The STPEEK specimens’ strength was slightly lower. The large volume of tert-butyl groups impeded the close packing of molecular chains to some extent. The STDPEEK specimens exhibited a lower strength and were relatively soft. However, the STDPEEK specimens exhibited a better compressive strength. It was explained that the larger steric hindrance restricted the arrangement of molecular chains.

Based on the aforementioned analysis findings, the following microscopic explanations were summarized and proposed: with the increase in rigid segments and the volume of the side groups, the steric hindrance of rotation within the main chains and the entanglement between molecular chains increased. This greatly restricted the internal rotation or bending of the molecular chains, which increased the intramolecular rotational energy barrier. Molecular rearrangement limited the slip of the SPEEK polymers and showed greater shear resistance at the macroscopic level [47,48,49,50].

## 4. Conclusions

In this study, the SPEEK specimens with different chemical structures were obtained through monomer polymerization and vacuum furnace to enhance lubricity and wear resistance in aqueous lubrication. Comparing three chemical structures, it was evident that the molecular structure was important to the tribological and mechanical properties. The conclusions drawn are as follows:
The appropriate introduction of the $–{\mathrm{SO}}_{3}^{-}$ groups can significantly reduce friction and wear through hydration lubrication and slightly enhance mechanical properties. The friction coefficients of SMPEEK, STPEEK, and STDPEEK were significantly reduced by 48%, 40%, and 30%, respectively, and the wear rates were correspondingly reduced by 58%, 51%, and 59%, respectively.The volume of side groups and rigid segments had a significant influence on tribological properties. It was described that the rotation and relaxation of molecular chains were influenced by the restriction of steric hindrance and entanglement between molecular chains.The SMPEEK exhibited superior anti-friction and anti-wear performance among the three chemical structures. The friction coefficient of SMPEEK could reach a value of 0.044 when subjected to a normal load of 15 N and a sliding speed of 50 mm/s under 3 wt% NaCl aqueous solution, and the wear rate was as low as 10^−8^ mm^3^/(N m). In addition, stability and low-cost monomers provided the premise for industrial application. In the future research, the tribological and mechanical properties can be further improved by fillers to meet the requirements of extreme working conditions. It indicated that SPEEK had great potential and advantages in water-lubricated bearings.


## Figures and Tables

**Figure 1 polymers-16-00998-f001:**
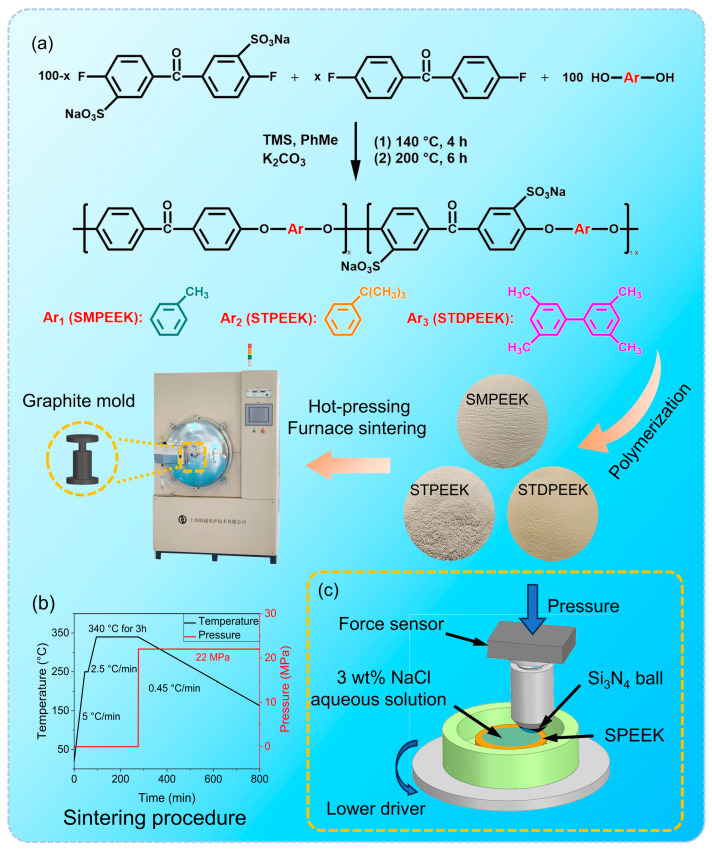
Preparation of the SPEEK and as-made pure PEEK specimens with three chemical structures: (**a**) reaction equations and processes; (**b**) sinter procedure; and (**c**) schematic of the working principle of the tribometer.

**Figure 2 polymers-16-00998-f002:**
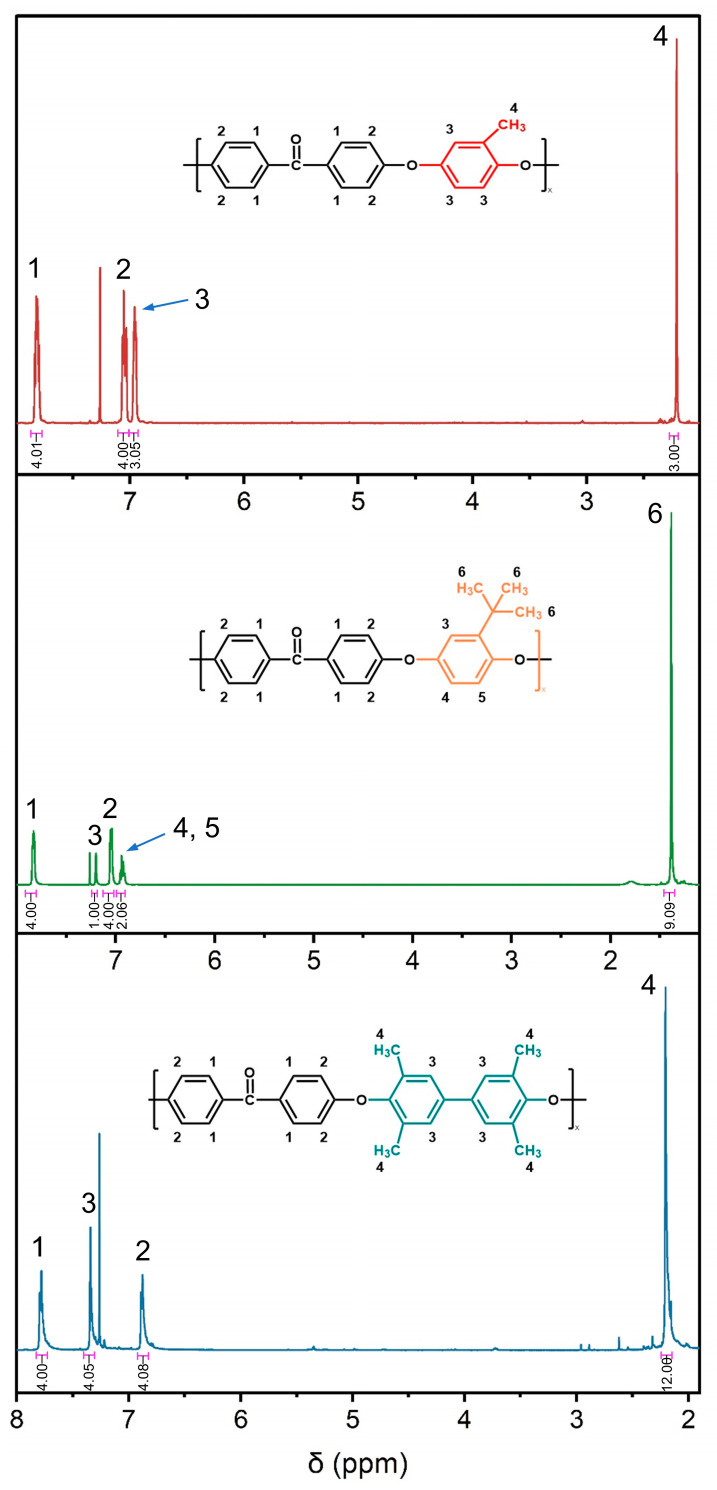
The ^1^H NMR spectra of SMPEEK, STPEEK, and STDPEEK.

**Figure 3 polymers-16-00998-f003:**
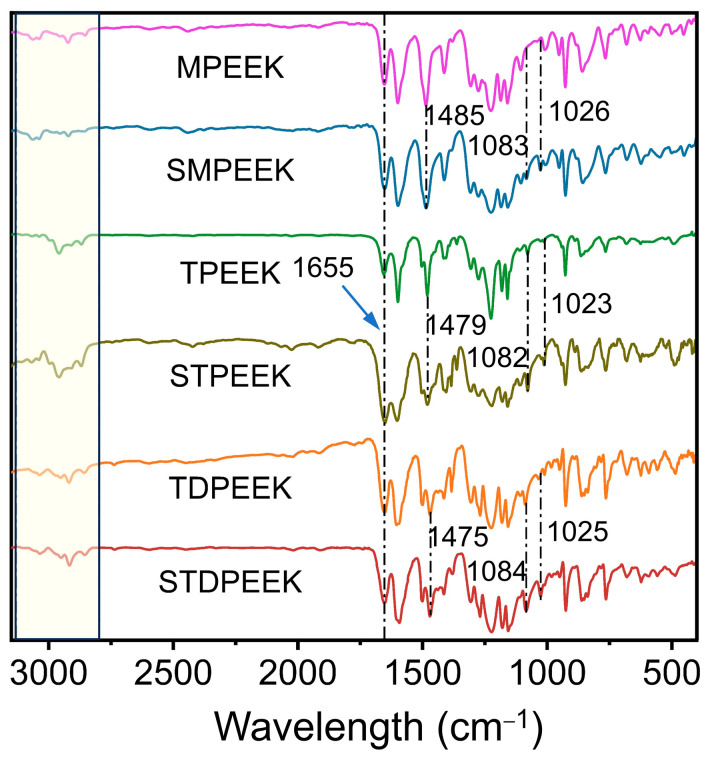
The FTIR spectra of SPEEK and the corresponding pure PEEK.

**Figure 4 polymers-16-00998-f004:**
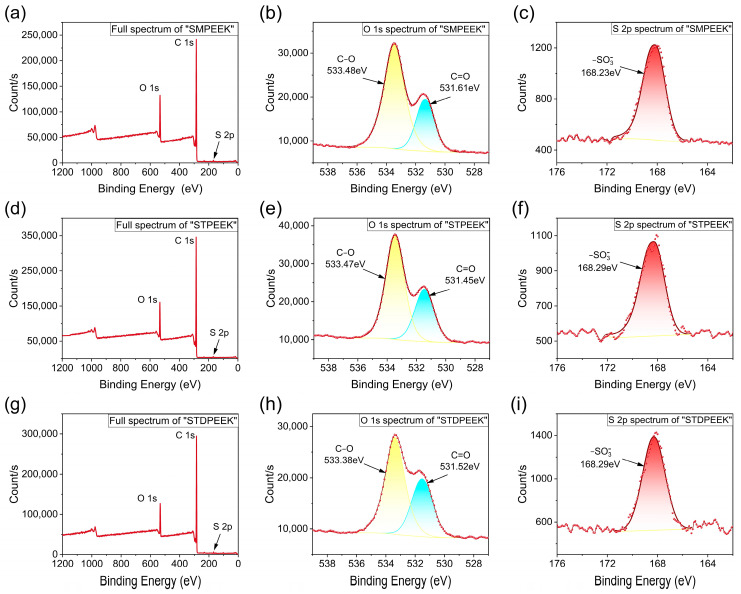
Full XPS, O1s, and S2p spectra of (**a**–**c**) SMPEEK, (**d**–**f**) STPEEK, and (**g**–**i**) STDPEEK.

**Figure 5 polymers-16-00998-f005:**
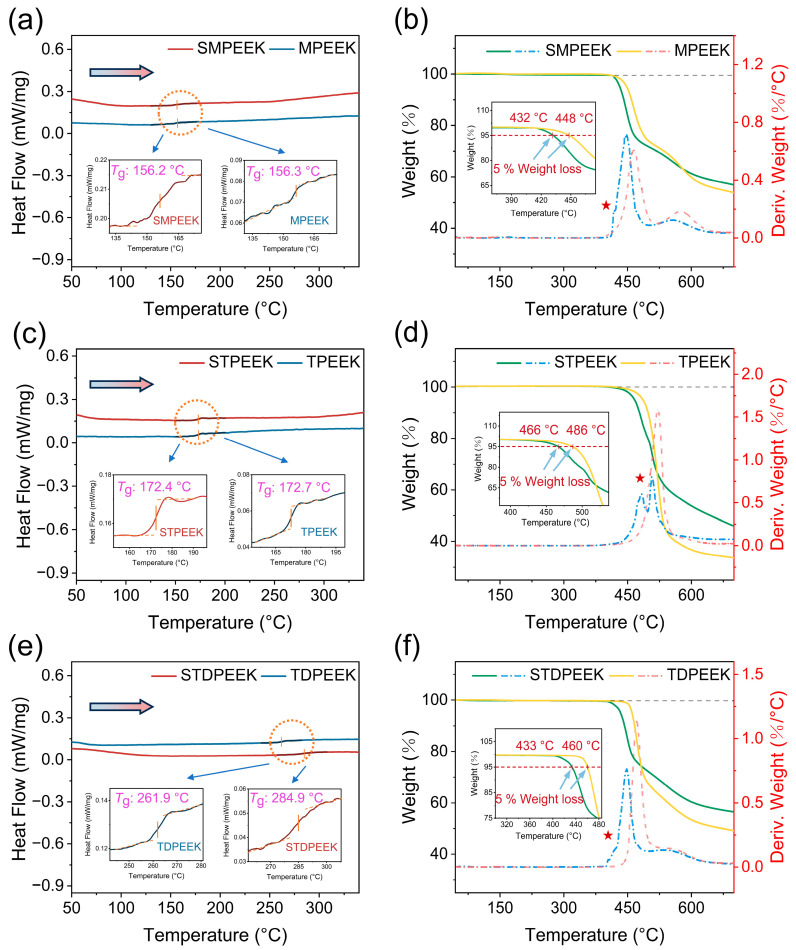
DSC and TGA profiles of (**a**,**b**) SMPEEK, (**c**,**d**) STPEEK, and (**e**,**f**) STDPEEK and their corresponding as-made pure PEEK.

**Figure 6 polymers-16-00998-f006:**
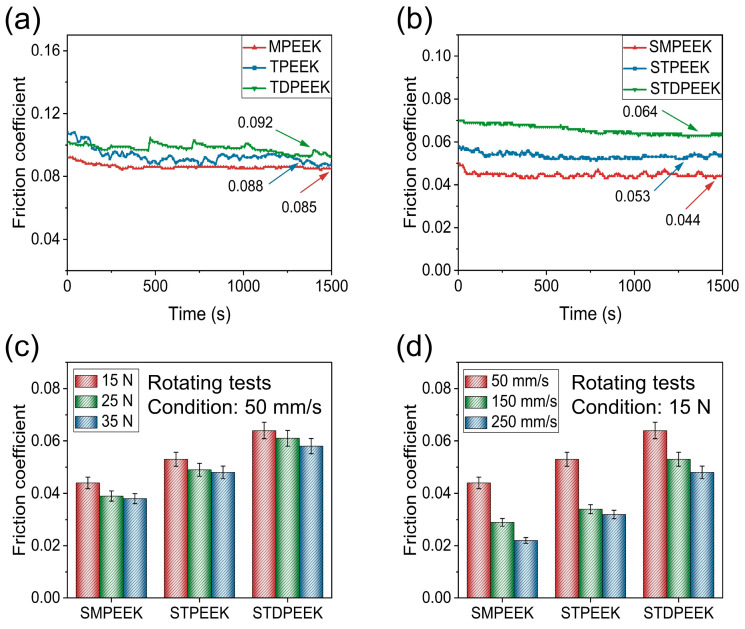
Friction coefficient of SPEEK and corresponding as-made pure PEEK with three chemical structures in aqueous lubrication through the rotating tribometer: (**a**) as-made pure PEEK, (**b**) SPEEK, and SPEEK at different (**c**) loads and (**d**) sliding speeds.

**Figure 7 polymers-16-00998-f007:**
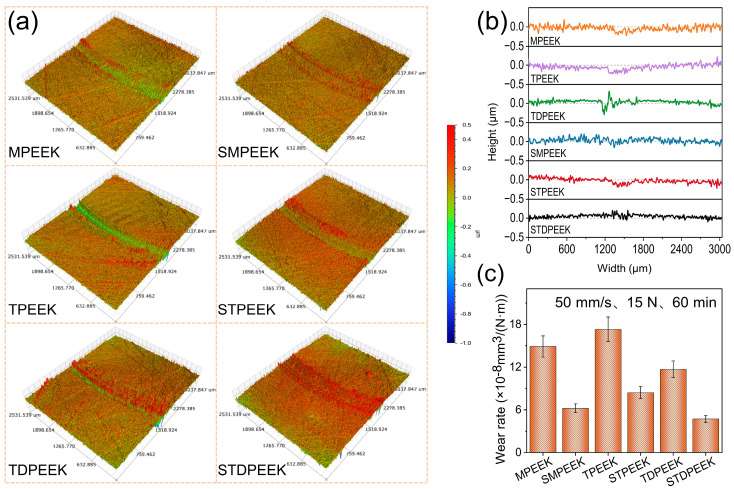
(**a**) Wear surface topography of SPEEK and as-made pure PEEK after rotating friction tests, and (**b**) their profile curves along the direction perpendicular to the velocity and (**c**) wear rate of the SPEEK and as-made pure PEEK with three chemical structures.

**Figure 8 polymers-16-00998-f008:**
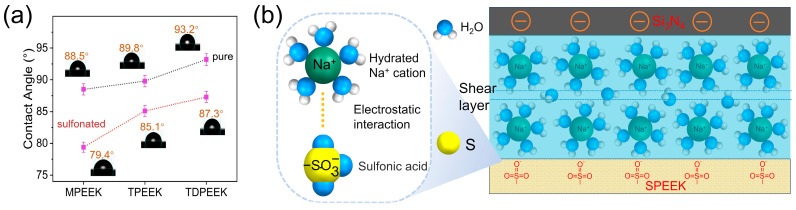
(**a**) Static contact angles of SPEEK and the corresponding pure PEEK. (**b**) Model of the hydration lubrication.

**Figure 9 polymers-16-00998-f009:**
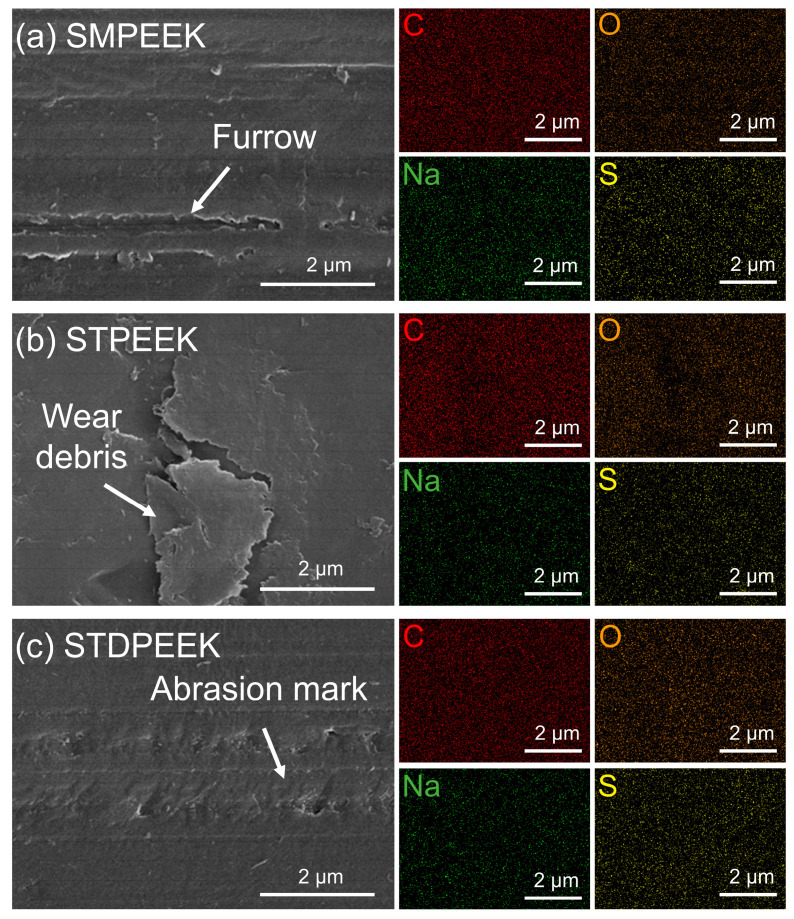
SEM and EDS images of the wear surface of (**a**) SMPEEK, (**b**) STPEEK, and (**c**) STDPEEK after a rotating test for 60 min.

**Table 1 polymers-16-00998-t001:** Amount of each monomer added in polymerization reaction.

Polymer	Monomer 1 (g)	Monomer 2 (g)	Monomer 3 (g)
MPEEK	0	21.8	12.4
TPEEK	0	21.8	16.6
TDPEEK	0	21.8	24.2
SMPEEK	2.1	20.7	12.4
STPEEK	3.2	20.2	16.6
STDPEEK	3.2	20.2	24.2

**Table 2 polymers-16-00998-t002:** Relevant characterization of the SPEEK powders.

Polymer	*M* _n_	*M* _w_	*D* _s_	*D* _50_
SMPEEK	6545	12,785	1.4	39
STEEK	11,761	20,689	2.6	53
STDPEEK	10,773	20,439	2.5	28

**Table 3 polymers-16-00998-t003:** The related mechanical parameters of the as-made pure PEEK and SPEEK specimens.

Polymer	Young’s Modulus (GPa)		Shore Hardness (D)		Compressive Strength (MPa)	
	Mean Value	Standard Deviation	Mean Value	Standard Deviation	Mean Value	Standard Deviation
MPEEK	4.4	0.2	85	2	125	3.5
TPEEK	4.1	0.3	84	5	114	4.0
TDPEEK	3.1	0.2	83	3	102	2.7
SMPEEK	4.5	0.1	84	1	132	2.5
STEEK	4.1	0.2	83	4	118	2.8
STDPEEK	3.6	0.1	78	3	146	2.3

## Data Availability

The raw/processed data generated in this work are available upon request from the corresponding author.

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
