# Peer review of "Tribological Behavior of Sulfonated Polyether Ether Ketone with Three Different Chemical Structures under Water Lubrication"

_polymers, 2024, doi:10.3390/polym16070998_

Round 1

Reviewer 1 Report

Comments and Suggestions for Authors

Tao Hu, Xiaohong Yi, and Fenghua Li et al., in their manuscript entitled "Tribological behavior of sulfonated PEEK with three different chemical structures under water lubrication", synthesized and investigated sulfonated PEEK (SPEEK) sintered specimens with varying side groups (methyl and tert-butyl) and rigid segments (biphenyl). They found that the friction coefficient and wear rate of SPEEK were significantly reduced by 30% and 50%. However, some of the limitations listed below were observed:

  1. The manuscript contains many typos and spelling mistakes. They should be corrected.
  2. The literature review about sulfonated PEEK is weak. The authors should add more information about the available sulfonated PEEK composites, especially from the point of view of mechanical and tribological properties.
  3. The authors should better illustrate the novelty of their work in the abstract and introduction.
  4. The authors should add the number of the tested samples for each test in sections “2.4. Characterization and 2.5. Tribological and mechanical tests”.
  5. The resolution of Figure 2 and Figure 7 should be improved. 
  6. The authors should refer to the references used in the explanation of the thermal properties of the prepared composites (3.1. Thermal properties analysis).
  7. The determination of the Tg values based on the results in Figure 5 is not clear. 
  8. The standard deviations for the mechanical properties in Table 3 should be added.
  9.  In the conclusion, the authors should add a brief about the prospects of their SPEEK.
  10. The authors should add DOIs for the references.
Comments on the Quality of English Language

Minor editing of the English language is required.

Reviewer 2 Report

Comments and Suggestions for Authors

In this paper sulfonated PEEK (SPEEK) thermoplastic polymers with methyl and tert-butyl side groups as well as biphenyl segments were synthesized and their tribological properties investigated for application in water-lubricated bearings. The synthesized polymers were found to exhibit considerably lower friction coefficients and wear rates compared to pristine PEEK. The paper is recommended for publication. A few specific comments/suggestions follow:

1. In the paragraph between lines 58-73 the advantages and benefits of having a negative electrically modified  PEEK should be briefly outlined. This will also emphasize the motivation of the present study.

2. In Table 1: "STEEK" should be "STPEEK".

3. Lines 115-117: the washing procedure of the powders should be described a bit better.

4. In the chemical reaction described in Figure 1(a) how was the Fluorine removed?

5. Lines 224-226: 'The smaller particle size..." the particle size was smaller compared to what? please specify.

6. Line 274: “The abrasive produced by wear caused numerical fluctuations.” The meaning of this sentence is not clear.

7. Please improve the resolution of the AFM images in Figure 7(a). The numbers are not distinguished.

Comments on the Quality of English Language

Minor editing of English language required.

Reviewer 3 Report

Comments and Suggestions for Authors Comments for polymers-2910765     Title: Tribological behavior of sulfonated PEEK with three different chemical structures under water lubrication

The paper focused on enhancing the tribological properties of polyetheretherketone (PEEK) as a water-lubricated bearing material within the shipbuilding industry. The study synthesized sulfonated PEEK (SPEEK) sintered specimens with varying side groups (methyl and tert-butyl) and rigid segments (biphenyl) to compare their performance. The results showed that the introduction of sulfonic acid groups in SPEEK significantly reduced the friction coefficient and wear rate by at least 30% and 50%, respectively, compared to as-made PEEK. The study suggested that the lower steric hindrance and entanglement between molecular chains, particularly in SPEEK with methyl side groups, contributed to its superior friction behavior, positioning it as a promising option for water-lubricated bearings.

After a thorough peer-review process, I am pleased to inform you that the paper has been deemed intriguing and has the potential for publication following significant "Major revisions".

Major comments

1. Provide more detailed information on the synthesis process of SPEEK specimens in the methodology section.

2. Its more beneficial to provide a discussion on the potential effects of temperature and pressure variations on the tribological properties of SPEEK.

3. Conduct a more in-depth analysis of the chemical interactions between the sulfonic acid groups and water molecules in the lubrication process.

4. Please address the cost-effectiveness and scalability of producing SPEEK for industrial applications in the conclusion section.

Round 2

Reviewer 3 Report

Comments and Suggestions for Authors

Accept as is